# Creation of acoustic vortex knots

Hongkuan Zhang [1,4], Weixuan Zhang [2,4], Yunhong Liao [1,4], Xiaoming Zhou [1✉], Junfei Li [3], Gengkai Hu[1] & Xiangdong Zhang[2✉]

Knots and links have been conjectured to play a fundamental role in a wide range of scientific fields. Recently, tying isolated vortex knots in the complex optical field has been realized. However, how to construct the acoustic vortex knot is still an unknown problem. Here we propose theoretically and demonstrate experimentally the creation of acoustic vortex knots using metamaterials, with decoupled modulation of transmitted phase and amplitude. Based on the numerical simulation, we find that the knot function can be embedded into the acoustic field by designed metamaterials with only 24 × 24 pixels. Furthermore, using the optimized metamaterials, the acoustic fields with Hopf link and trefoil knot vortex lines have been observed experimentally.

[1] Key Laboratory of Dynamics and Control of Flight Vehicle, Ministry of Education and School of Aerospace Engineering, Beijing Institute of Technology, 100081 Beijing, China. [2] Key Laboratory of Advanced Optoelectronic Quantum Architecture and Measurements of Ministry of Education, School of Physics, Beijing Institute of Technology, 100081 Beijing, China. [3] Department of Electrical and Computer Engineering, Duke University, Durham, NC 27708, USA. [4] These authors contributed equally: Hongkuan Zhang, Weixuan Zhang, Yunhong Liao. ✉email: zhxming@bit.edu.cn; zhangxd@bit.edu.cn

The exploration of knot physics has become one of the most fascinating frontiers in recent years, because its complex topology can always play an important role in the physical and life sciences[1–14]. At present, the knot-type topological structures have been realized in various systems, including plasmas[2], quantum and classical fluids[3–5], quantum and classical field theory[6–9], and liquid crystals[10,11]. In electromagnetic fields, the isolated optical vortex[15–23] and polarization knots[24,25] have also been engineered and observed, which are considered to have potential applications to confine cold atoms with complex spatial topologies. However, it is still unknown whether the acoustic wave can also possess the same topological properties.

Metamaterials that consist of manmade elementary building blocks can exhibit acoustic characteristics beyond the properties of natural materials[26–28]. Acoustic metamaterials comprising locally resonant cells were shown to exhibit anomalous reflection/refraction[29,30], acoustic cloaking[31,32], sub-diffraction-limited imaging[33], topological propagation[34–37], etc. Especially, an ultrathin metamaterial with a planar profile, also known as the metasurface[38–40], was able to offer the convenient wavefront modulation, suggesting a novel approach to acoustic holograms[41,42]. Recent investigations have shown that the orbital angular momentum of sound fields with straight vortex lines can be efficiently engineered by using a metamaterial-based phased array[43]. It is therefore relevant to ask whether the acoustic fields with knotted vortex lines can be generated by metamaterial-based three-dimensional (3D) holographic projection.

In this work, we design theoretically and fabricate experimentally a metasurface to create acoustic vortex knots. The knot function, derived from abstract functions with braided zero-value lines, is embedded into a propagating acoustic beam by a suitably engineered metasurface. Based on the finite element method, we numerically demonstrate that both Hopf link and trefoil knot vortex lines can be generated by an initial wavefront with only $24 \times 24$ sub-pixels accompanied with suitable phase and amplitude distributions. The required phase and amplitude patterns can be physically realized using unit cells with decoupled modulation of transmitted phase and amplitude. The experimental measurement of acoustic vortex knots is carried out by an acoustic pulse-echo scanning system. Both Hopf link and trefoil knot vortex lines at wavelength scale have been observed. These acoustic vortex knots possess potential applications in the field of particle trapping with the real space topology.

## Results

### The general scheme to create acoustic vortex knots.
We begin with constructing the complex-valued mathematic functions with knotted zero lines in 3D spaces. Here the focus is placed on the generation of torus knots. Based on the knot theory, the function with knotted zeroes can be realized by devising complex functions with zero lines on a periodic braid[22]. In this case, the helical zero-value braid in the complex plane is expressed as:

$$Q_{\text{helix}}(u, v) = u^2 - v^n, \quad (1)$$

where $u$ and $v$ are complex variables and $n$ is the number of repeats of the basic crossing sequence. Then we convert the two-dimensional complex polynomial $Q_{\text{helix}}(u, v)$ into 3D real spaces with suitable coordinate transformations. In this case, the Hopf link $f_{\text{Hopf}}(x, y, z)$ and trefoil knot $f_{\text{trefoil}}(x, y, z)$ algebraic functions can be achieved (see Note 1 in Supplementary Information for details). Based on the holographic projection scheme, the knotted acoustic field can be generated by the source plane with the initial wavefront embedded with knot functions (at the particular plane $z = z_0$). The initial wavefront plane of the acoustic fields with the ability to create Hopf link and trefoil knot vortex

loops can be described by:

$$\psi_{\text{Hopf}}(x, y) = f_{\text{Hopf}}\left(x, y, z_{\text{Hopf}}\right) \exp\left(-\frac{x^2 + y^2}{2W_{\text{Hopf}}^2}\right) \text{ and } \quad (2)$$

$$\psi_{\text{trefoil}}(x, y) = f_{\text{trefoil}}\left(x, y, z_{\text{trefoil}}\right) \exp\left(-\frac{x^2 + y^2}{2W_{\text{trefoil}}^2}\right), \quad (3)$$

where the parameters $W_{\text{Hopf}}$ ($W_{\text{trefoil}}$) and $z_{\text{Hopf}}$ ($z_{\text{trefoil}}$) represent the normalized width of the Gaussian profile and the selected position of transverse-plane embedded on the initial wavefront for the Hopf link (trefoil knot) vortex field, respectively (see Note 2 in Supplementary Information for detailed values of these parameters).

With the realization of optical vortex knots for nearly a decade, the method to construct acoustic vortex knots remains still lacking. It is probably owing to the following challenges faced while extending the optical schemes to acoustic versions. (i) The overall size of acoustic devices should be extremely large. As many previous works have demonstrated, the construction of optical vortex knots can be realized based on the diffractive holographic scheme with phase-only hologram[22] (see Note 3 in Supplementary Information for details). To fulfill the high diffraction efficiency, the designed hologram should possess a large number of pixels (about $300 \times 300$) for the smooth phase modulation induced by the blazed grating. However, the acoustic analog that shares a similar number of pixels (even each pixel is at subwavelength scale) would be unacceptably bulky. For example, the transverse size of the sample may be up to $6\,\text{m} \times 6\,\text{m}$ for an audible frequency $6\,\text{kHz}$, given that the pixel size is of $1/3$ operating wavelength. (ii) Except for the fabrication challenge of bulky samples, the creation of acoustic vortex knots in a giant space (around $6\,\text{m} \times 6\,\text{m} \times 6\,\text{m}$) raises a great difficulty for experimental characterization.

To tackle these problems, a hologram-based method with both phase and amplitude control is adopted to project the acoustic vortex knots with a metasurface. The transmission phase–amplitude distributions of holograms needed for encrypting the non-trivial topological information of the generated 3D acoustic field are obtained based on the algebraic topology (from Eqs. (2) and (3)). To facilitate the sample fabrication and testing, the optimization is performed to determine the minimum number of cells required. According to numerical analyses, the minimum number of pixels is found as $12 \times 12$ for the Hopf link and $16 \times 16$ for the trefoil knot (see Note 4 in Supplementary Information for details); below this lower limit, the topological phase transition of the vortex line (broken and reconnection) would appear such that the knot and link are destroyed. In principle, the larger the number of pixels is, the better is the knotted vortex performance. As the cell number decreases, the Hopf link and trefoil knot vortex lines deform in size and shape, yet maintain topologically invariant. The topology, here, means that knots are equivalent to each other if one can be transformed into the other via the continuous deformation without cutting the lines or permitting the lines to pass through itself[3–11]. In our study, a moderate pixel number $24 \times 24$ that lies above the lower limit is chosen, which is adequate to our needs for the vortex knot demonstration. Figure 1a, b (Fig. 1c, d) present the amplitude and phase distribution of the hologram with $24 \times 24$ pixels to generate the Hopf link (trefoil knot) vortex line, respectively. Note that, even with roughly discretized phase–amplitude distributions induced by the hologram, it is still able to generate the Hopf link and trefoil knot vortex lines thanks to the topological robustness of knotted fields.

The required transmission phase and amplitude at each pixel can be physically realized by using a metamaterial element

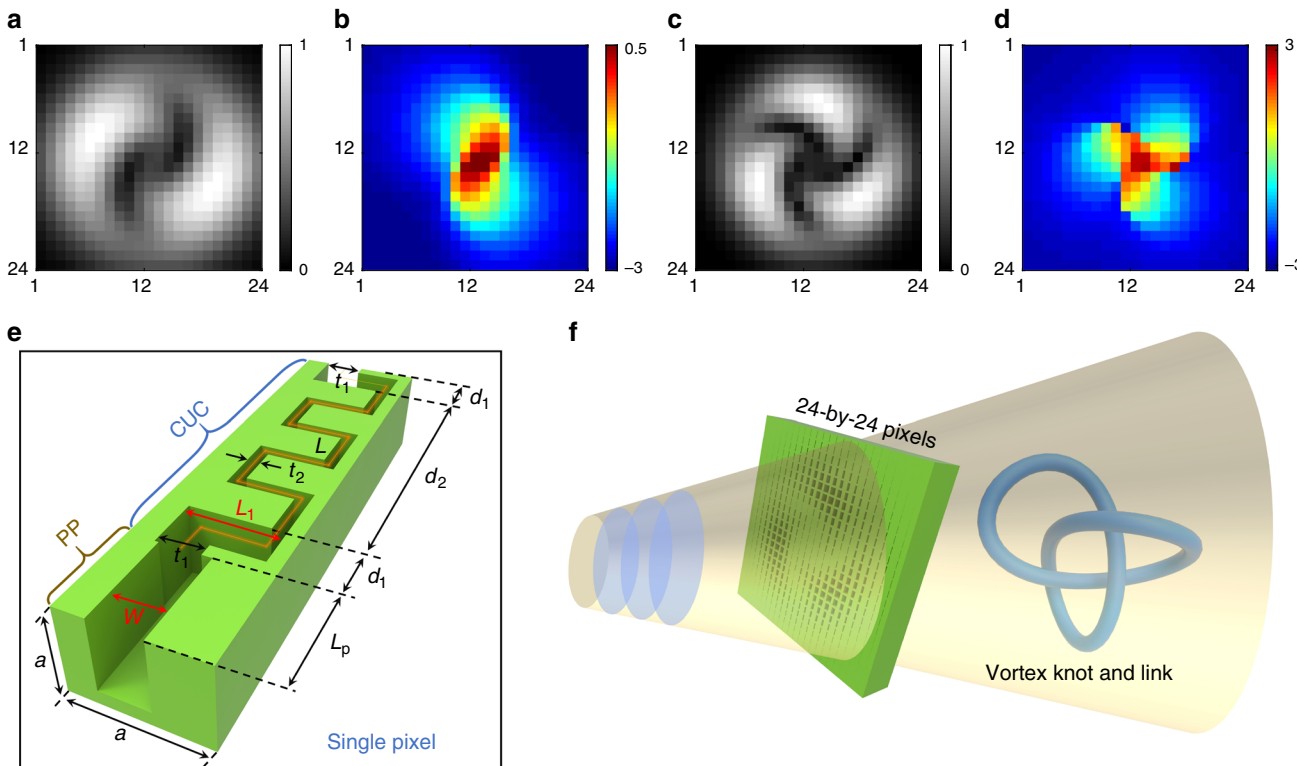

**Fig. 1 The schematic of generating acoustic vortex knots with metamaterials. a**, **b** plot the amplitude and phase distribution of the initial wavefront with 24-by-24 pixels to generate the Hopf link vortex line. **c**, **d** present the distribution of amplitude and phase on the initial wavefront with 24-by-24 pixels for the generation of the trefoil knot vortex line. **e** The structure of a single unit cell. The operation wavelength is $\lambda_0 = 57.17$ mm (6000 Hz). The width of the structure is $a = \lambda_0/3 = 19.07$ mm. Here geometric parameters of CUC boundary layers are chosen as $d_1 = 8.17$ mm, $t_1 = 6.17$ mm, and $L_1 = 13.3$ mm. And geometric parameters of the CUC center layer are $d_2 = 43.66$ mm and $t_2 = 2$ mm. The relative phase shift of the transmitted field can be tuned from 0 to $2\pi$ with $L$ (the orange line) changing from 2 to 16.05 mm. The length of PP is $L_p = \lambda_0/4 = 28.585$ mm. The width of PP is $W$. **f** The schematic of creating acoustic vortex knots by using 2D metamaterial-based hologram with both phase and amplitude control.

composed of coating unit cell (CUC) and perforated panels (PP) under the plane wave illumination[44,45], as shown in Fig. 1e. The CUC consists of three layers of suitable sized labyrinthine structures. It is nearly acoustically transparent if the viscous loss of air is neglected, while offering the independent control over the full range of the transmission phase from 0 to $2\pi$ by changing pipeline length ($L$, the orange line marked in Fig. 1e) of the central labyrinthine layer. The PP structure provides a precise amplitude control by appropriately tuning the width ($W$) of holes, yet along with a nearly constant phase shifting. A compound having CUC and PP has realized some interesting phenomena, including acoustic holography[44] and analog computing[45]. However, different from previous designs, our holograms are required to modulate simultaneously the transmission phase and amplitude in wide ranges. In this case, the hologram works near the resonant frequency of PPs, where the viscous loss of air must be considered. The viscous loss of air is quantified by adding an imaginary part to the sound velocity, which is determined by matching simulated and measured transmission of a single unit cell, and the fitted sound velocity is given by $c_0 = 343.2 \times (1 + i0.012)$. The schematic diagram of using metamaterial-based holograms to generate the acoustic vortex knots is shown in Fig. 1f. The metasurface is built by $24 \times 24$ cells, capable of generating prescribed phase and amplitude distributions (as shown in Fig. 1a–d) at the output surface, where a single metamaterial cell that acts as one pixel is designed based on the optimization algorithm in COMSOL (Coordinate Search Algorithm). The knotted acoustic fields are expected in the near-field region behind the metamaterial.

**Experimental realization of the acoustic vortex knots**. The schematic diagram of the experimental set-up to demonstrate the vortex knot performance is shown in Fig. 2c. It is worth noting that, although the proposed scheme for the creation of acoustic vortex knots is not limited to a specific range of frequencies, 6 kHz is chosen for vortex field demonstration in the experiment within the limitation of the measurement area of the scanning stage (50 cm × 50 cm) and the fabrication capability of the 3D printing technique.

The metamaterials are fabricated with photopolymer via stereolithography apparatus (SLA) 3D printing technique. Details of sample fabrication are provided in the "Methods" section. The side and front views of the photographic image of the fabricated metamaterials, which sustain Hopf link and trefoil knot vortex loops, are shown in Fig. 2a, b, respectively. The sample is about $45 \times 45$ cm$^2$ in the transverse plane and the thickness is about 8.86 cm. We note that different geometric parameters of each pixel correspond to the spatial-dependent phase and amplitude manipulation. A modulated Gaussian pulse sound with a center frequency of 6 kHz is generated by a loudspeaker and is incident on the sample. The sound fields projected on the transmission side of the sample are measured by a tri-axis scanning stage, at a distance from which is an anechoic wall used for reducing the sound reflection. Details of the measurement system are provided in the "Methods" section. Based on such an experimental set-up, the acoustic vortex knots generated by the metamaterials can be characterized.

Acoustic vortex links and knots are the patterns formed by the dark points, at which theoretically the pressure amplitude is zero. Meanwhile, the dark points correspond to vortex points with phase singularities. The experimental results of ideal "zero" pressure

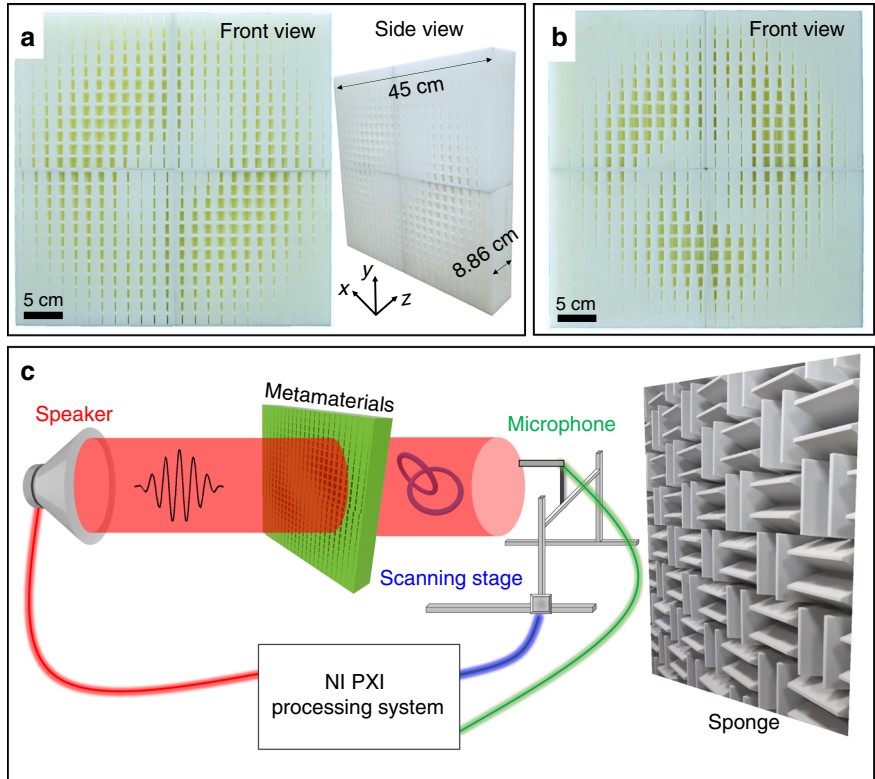

**Fig. 2 The fabricated sample and experimental set-up. a**, **b** show the photograph of the fabricated metamaterials sustaining Hopf link and trefoil knot vortex loops, respectively. Both samples are about $45 \times 45 \, cm^2$ in the transverse plane and the thickness is about 8.86 cm. **c** The schematic diagram of the experimental set-up.

amplitude are not available because the background noises, though largely reduced in experimental tests, cannot be fully removed. For this reason, the experimental dark points are recognized by seeking the local minima of pressure amplitude contour. The measured results for characterizing the two fabricated metamaterials at 6 kHz are presented in Fig. 3. At first, we consider the metamaterial sustaining Hopf link vortex line. The measured amplitude and phase distributions of the acoustic field in different transverse planes ($z = 2$, 7, 12, 17, 22, 27, 32, and 37 cm) are shown in Fig. 3a. Each plane possesses four dark points (marked by white dots), which are local minima of acoustic pressure contour. In a path circulating these dark points, the acoustic pressure phase undergoes one complete $2\pi$ rad cycle, forming the phase vortex pattern. Both sets of results and their excellent agreement show a clear demonstration of acoustic vortex points. Clearly, locations of four dark points evolve with acoustic wave propagation. We note that two pairs of the initially separated dark points recombine, indicating the formation of the closed loop. Connecting these dark points at different transverse planes, an isolated Hopf link vortex loop can be formed, as shown in Fig. 3b. The measured Hopf link vortex loop is about 20 cm × 20 cm in the $x$–$y$ plane, and the longitudinal length is about 40 cm.

Similar to the condition of the Hopf link vortex line, we measure the image of amplitude and phase distributions at different transverse planes ($z = 2$, 7, 12, 17, 22, 27, 32, and 37 cm) for the metamaterial producing the trefoil knot vortex line, as shown in Fig. 3d. It is clearly shown that there are six dark points (marked by white dots) in each plane, where the phase shows the cyclic change around each dark point, establishing the singular phase nature of the intensity nodes. In this case, an isolated vortex trefoil knot can be observed by connecting these dark points, as shown in Fig. 3e. The measured trefoil knot vortex loop is about 25 cm × 25 cm in the $x$–$y$ plane, and the longitudinal length is also about 40 cm. For comparison, the simulated Hopf link and trefoil

knot vortex loops are shown in Fig. 3c, f from an identical viewpoint to the measured ones. There is a small discrepancy between the simulation and experiment results due to the fabrication error and environmental noise. However, in spite of the slight change of the vortex loop in size and shape for both sets of results, the topological features of the Hopf link and trefoil knot vortex lines are preserved.

## Discussion

To highlight the novelty of our work, the difference between the creation of knotted vortex line for the acoustic and electromagnetic field is generalized below. In electromagnetic fields, the behavior of a monochromatic light beam propagating along the $z$-direction should be described by the vector Helmholtz equation. For the transverse paraxial case, the optical vortex knot has been experimentally constructed in the laser beam[16–18,20–23] based on the far-field holographic scheme, where the algebraic knot function at $z = 0$ plane [$f_{Hopf}(x, y, z = 0)$ or $f_{trefoil}(x, y, z = 0)$] is embedded into the waist of a Gaussian beam. In another case, it has been proposed theoretically that the longitudinal polarization component of light with the knotted vortex line can be formed by non-paraxially focusing a subwavelength optical beam[19]. The generated vortex knots can then be identified by examining the distribution of both intensity and phase for the longitudinal polarization of the light field. It is worth emphasizing that the formation of knotted vortices for optical fields was based on the phase-only hologram. As a result, the optical hologram should possess a large number of pixels (about 300 × 300) to achieve a high diffraction efficiency.

By contrast, the acoustic field is characterized by a scalar pressure. To generate the unbroken vortex knot in the transmission region, the hologram with both phase and amplitude control is adopted in the present work, in which an algebraic knot function at the $z = z_{Hopf}$ (or

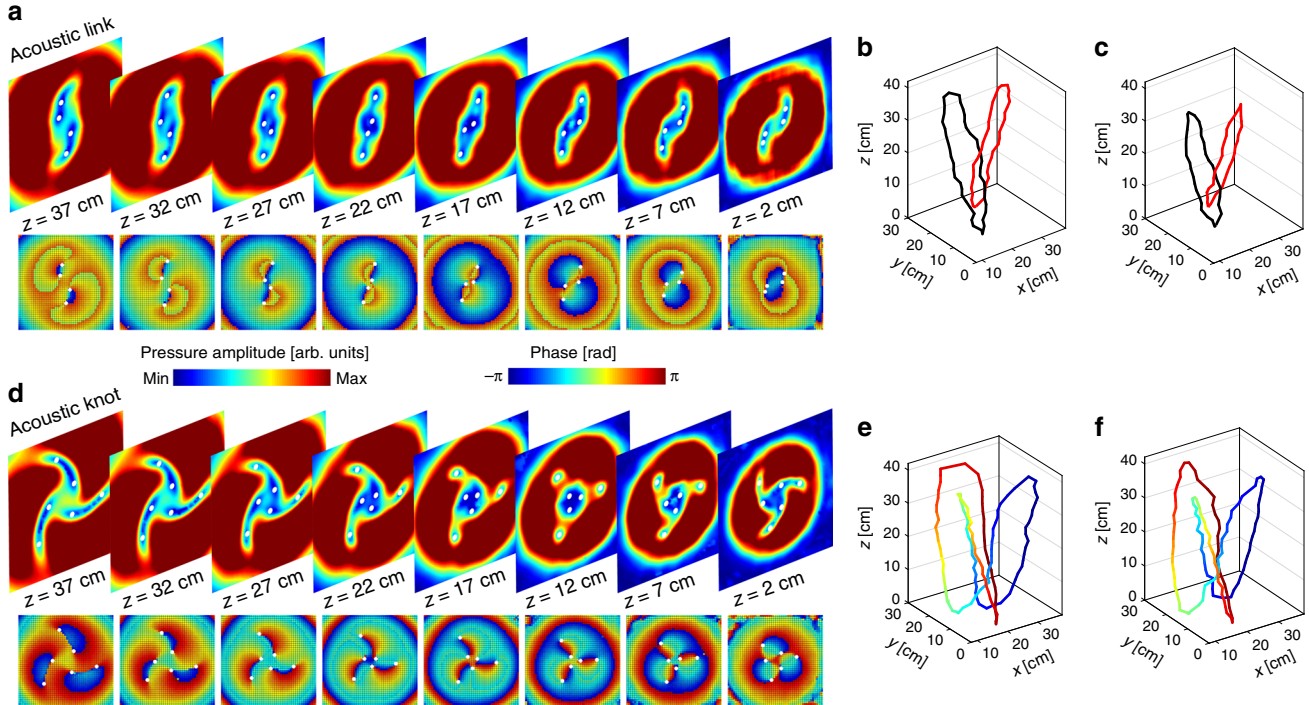

**Fig. 3 The measured acoustic fields with Hopf link and trefoil knot vortex lines.** The left charts show the amplitude and phase distributions at different $z$-plane for **a** Hopf link and **d** trefoil knot. The white dots possessing the singular phase represent the vortices. The right charts plot the experimental Hopf link (**b**) and trefoil knot (**e**) as well as simulation ones (**c**, **f**), where the color marked on the vortex lines is used to distinguish the knotted structure with overlapped regions.

$z = z_{\text{trefoil}}$) plane [$f_{\text{Hopf}}(x, y, z = z_{\text{Hopf}})$ or $f_{\text{trefoil}}(x, y, z = z_{\text{trefoil}})$] is embedded into the initial wavefront at the aperture. In this case, a fewer number of pixels are needed comparing with the phase-only aperture for the optical condition. In the present study, acoustic vortex knots are created by a compact metasurface with only $24 \times 24$ pixels. In this case, the ratio of the lateral size of the sample to the operating wavelength is much less than that of a phase-only optical hologram. To summarize, the present work not only provides the experimental demonstration for acoustic links and knots but also indicates the possibility to create linked and knotted wave fields in the near-field regime using a compact aperture with fewer pixels, which is believed to be inspirational for other physical fields.

In addition, our work may promote the development of using acoustic vortex knots for particle-trapping applications. It has been demonstrated that the acoustic nodes, i.e., low amplitude pockets surrounded by high pressure regions, could be used for trapping a single particle[46–49]. Moreover, acoustic node lines created by standing waves have also been proven to be used for particle manipulations[50–52]. Distinct from the previous node line pattern, the present work demonstrates the 3D topology configuration of knotted node lines, holding the possibility of patterning biological and chemical particles in the 3D space. To realize the particle trapping near dark points, the magnitude of the acoustic pressure gradient, which relies on both the wave amplitude and frequency, need to be sufficiently large. Thereby, the vortex fields created at the ultrasonic environment with a high intensity (up to 40 kHz, driven around 15 Vpp) are expected to possess the enough pressure gradient for the particle trapping on knotted vortex lines in experiments[46–49].

In conclusion, we have designed theoretically and fabricated experimentally a type of acoustic metamaterial to create vortex knots. Based on the finite element method, we have proved that knot functions, which are derived from the algebraic topology,

can be embedded into the propagating acoustic fields by the metamaterial consisting of only $24 \times 24$ pixels. In experiments, the acoustic fields with both Hopf link and trefoil knot vortex lines have been observed by the 3D scanning system. The created acoustic vortex knots hold potential for applications in the field of exotic interaction between acoustic knots with matter, so as to construct various space topological structures of the trapped micro-particles without contact.

## Methods

**Experiments**. All samples are fabricated via the SLA 3D printing technique (0.1 mm in precision). The base material is photopolymer with a density of 1130 kg m$^{-3}$ and Young's modulus 2.65 GPa. The photopolymer structure is modeled as a rigid frame due to a high impedance mismatch between the material and air. The designed acoustic link or knot metamaterial consists of 576 subwavelength channels forming a $24 \times 24$ lattice. Without influencing test results, the metamaterial sample is divided into four smaller sections of equal size. Each section is fabricated by using 3D printing and then combined together into one entire sample. A loudspeaker is placed at a distance 1.5 m from the sample and generates airborne sound with the nearly plane wavefront upon impinging the sample. The speaker emits a Gaussian-modulated sinusoidal packet with a center frequency of 6.0 kHz. On the transmission side far from the sample is a high-quality anechoic wall used for minimizing the sound reflection. The transmitted acoustic signal measured by a microphone is recorded by NI PXIe data acquisition and processing system, and after the Fourier transform, the amplitude and phase of the acoustic wave of frequency 6.0 kHz are retrieved. A tri-axis stepping motor is used to move the microphone and scan the pressure field in a cubic region of 50 cm × 50 cm × 50 cm with a step size of 5 mm. The stepping motor controller and NI PXIe system are communicated through a laboratory-made LabView program.

## Data availability
The data that support the findings of this study are available from the corresponding authors upon reasonable request.

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

## Acknowledgements

This work was supported by the National key R & D Program of China under Grant No. 2017YFA0303800, National Natural Science Foundation of China (11872111, 11991030, 11991033, 11622215, 11572039, and 11632003), and 111 project (B16003).

## Author contributions

W.Z. provided the theoretical design and analysis. H.Z. and Y.L. performed the experiment with the help of J.L. under the supervision of X. Zhou and G.H. X. Zhang initiated and designed this research project.

## Competing interests

The authors declare no competing interests.
