## [Peer Review File · Nature Communications]

Reviewers' Comments:

Reviewer #1:

Remarks to the Author:

The paper presents theoretical and experimental realizations of acoustic knotted vortices. I think the experimental measurements of these fragile and intricate fields are enough to warrant publication. I am sure that people working on knots (being them optic, plasma, water flow...) will appreciate this achievement. The paper is nicely written, to the point and properly motivated. That is why I would recommend the paper for acceptance.

However, I would like more information on how the knots were detected once that the acoustic field was scanned. In the paper, the authors talk about "dark points", I imagine that these are points at which theoretically the amplitude is 0. But, I imagine that experimentally there are some details about how to get this. Do the authors look for minima of amplitude? In any case, I am not convinced by this method. Most of the papers on knotted vortices that I have seen look in the phase and check for vortices using algorithms based on the phase. No plots of the phase appear in the main paper.

Trapping of particles in acoustic vortices or shaped fields is described but no reference is provided, I would recommend these:

- Marzo, A., Seah, S. A., Drinkwater, B. W., Sahoo, D. R., Long, B., & Subramanian, S. (2015). Holographic acoustic elements for manipulation of levitated objects. *Nature communications*, 6, 8661.
- Baresch, D., Thomas, J. L., & Marchiano, R. (2016). Observation of a single-beam gradient force acoustical trap for elastic particles: acoustical tweezers. *Physical review letters*, 116(2), 024301.

Reviewer #2:

Remarks to the Author:

This paper reports on theoretically and experimentally determined acoustic vortex knots (a Hopf link and Trefoil knot) using a metamaterial aperture. Design of the metamaterial was guided by theoretical formulations from topology for the particular types of vortex link and knot examined. The metamaterial aperture consists of an arrangement of 24 x 24 units (pixels), that was fabricated using SLA 3D printing. Each unit cell was numerically optimized and consisted of a coating unit cell and perforated panels that provided modulation of both the magnitude and phase. Acoustic testing was performed, and the 2D scans of sound pressure magnitude were taken at multiple distances from the metamaterial aperture, with results showing a clear demonstration of an acoustic link and acoustic knot.

This paper is well-written, and clearly and concisely describes the work presented. While vortex waves and vortex knots have been examined extensively in other fields of science (such as optics), it has to this reviewer's knowledge never been experimentally demonstrated in acoustics before. As a result, this work represents a significant and novel contribution to the scientific literature on this topic. Furthermore, the authors were thorough in their analysis, utilizing both theoretical and numerical simulation and optimization techniques, and even performing component level experiments examining the viscous effects of air on their metamaterial design (as described in the supplementary section).

Although overall very good, several minor issues were identified with the paper which should be addressed:

- 1) Given that this appears to be the first paper to demonstrate the effect of an acoustic link and acoustic knot, the authors should add some additional emphasis of their analysis and results in context

of previous results in related fields, such as optics. In particular, the authors should comment on if there are any differences between their equations and measured results for acoustic (longitudinal) waves and previous results with electromagnetic (transverse) waves.

2) While the Methods section is overall very detailed, the authors should clarify if the metamaterial was 3D printed as one entire piece, or was printed in smaller sections and assembled. If it was printed in small sections, the authors should describe how they attached those different sections together.

3) A few minor typos were found that should be corrected:

- In the following text from p. 4: "For example, the transverse size of the sample may be up to 6m x 6m for an audible frequency 6 kHz, suppose that the pixel size is of 1/3 operating wavelength. (ii) Except the fabrication trouble of bulky samples..."

Here, "suppose" should be replaced with a different word, such as "given", and "Except the" should be "Except for the".

- In the last paragraph on p. 4, In "To enable of proposal experimentally feasible", "enable" should be "make". At the end of the same sentence, "24 multiplied by 24 pixels" should just read "24 x 24 pixels".

- In the first line of p. 6, "At the case" should be "For the case",

Once the issues listed above are addressed, I recommend that this manuscript is ready for publication in Nature Communications.

Reviewer #3:

Remarks to the Author:

In this article, Zhang et al describe the creation of acoustic knots--essentially acoustic pressure holograms with knot-like 3D shapes--by means of a planar metamaterial array whose units can independently manipulate amplitude and phase. While the metamaterial design is not novel (it was introduced by Tian et al. in their 2017 letter "Acoustic holography based on composite metasurface with decoupled modulation of phase and amplitude", Ref.[42] in this article), the application to knots seems to be. The topic of this article is interesting, and it has the potential to attract the attention of others in the field. However, the fact that some concepts are not clearly explained and the absence of a powerful demonstration of the usefulness of the phenomenon being investigated prompt me to think that this article might be better suited for a technical journal rather than a high impact multidisciplinary venue like Nature Communications.

The reasoning behind my statement is elaborated in the following.

1) Concern on novelty. While it is very interesting that the authors were able to create knotted regions of negative pressure, I can't help but wonder whether this is something that is easy to achieve given the design in Ref.[42]. Can the authors clarify if the novelty of their work goes beyond creating a 3D hologram of a knot? Or, how is what they are doing different than the holography demonstrated in Ref.[42]?

2) Concern on the applicability of knots for particle manipulation. It seems to me that the biggest achievement of this article is the creation of knot-like holograms, where the knots are essentially regions of minimum pressure. But is the pressure gradient between high- and low-pressure regions enough to concentrate particles at the knot regions? From [Memoli et al, Metamaterial bricks and quantization of meta-surfaces, Nature Communications, 2017], particle trapping can occur in low amplitude pockets surrounded by high pressure regions. But, in the current work, the low pressure

regions seem to be distributed and wide, rather than highly localized. Do the authors have any insight in this?

3) The article seems scientifically sound, but it lacks a field-opening demonstration that would justify publication in a high-profile journal like Nature Communications. Have the authors tried to demonstrate that their knots can indeed be used for complex 3D particle manipulation? In my opinion, a demonstration of this type would raise the article to a much higher level. Just speculating on the potential applicability of these knots, in my opinion, is not enough.

4) About topological protection of the knotted shapes. The authors mention in several points of the article that their knots are topologically-protected, and this is the reason why 24x24 cells are enough to see knot-like shapes. I found this argument very hand-wavy, and unsatisfactory. How does topology play a role here? If you lower the number of cells below 24x24, do you still see knotted shapes owing to this topological protection? When would this protection cease to be effective? I urge the authors to be clearer, and more rigorous, about this.

5) Level of detail. Most explanations in the article are not very clear; this stems from the fact that the authors do not provide a sufficient level of detail in the main article. Sure, the Supplementary Material (SM) helps. But the SM should not be used as a place to dump all the explanations, leaving the main article to become a superficial treatment of the topic.

- For example, the authors talk about optimization to design the metamaterial with 24x24 cells. But how is this optimization done? Why did the authors choose 24x24 cells? What happens if they choose more or less? This is not even clear from the SM.

- The geometry of a unit cell is described in the caption of Fig.1, and the authors mention the pipeline length L in both the main text and caption. However, from the figure, it is not really clear what L is. It is also not clear if all that matters is the path length or its shape.

- Why do the authors operate the metamaterial at a frequency of 6 kHz? There is absolutely no explanation on this point in the article. Did they choose the frequency arbitrarily? This demonstrates that the treatment is superficial and that the work is hardly reproducible the way it is at the moment.

- How are the 3D reconstructions of the knots in Fig.3 obtained? Are the authors tracking the minima of the pressure slices shown in the same figure? It seems to me that these pressure maps present entire regions with pressure close to 0. So, how do the authors select only two or four points per slice? Are they local minima? This is absolutely not clear from the text. Also, how is the trefoil knot colored in Fig.3b?

Minor suggestions

- The work is not badly written, but the article "the" is incorrectly used in various parts of the text. This should be fixed.

- The authors, in the main text, comment on the partial differences between numerical and experimental knots. However, these differences are very hard to see, since the two sets of knots are not plotted next to each other. I think the authors should add the numerical 3D knots to Fig.3, and plot both numerical and experimental knots from an identical viewpoint. Only then, it would be possible for the reader to compare them directly.

Article: “Creation of acoustic vortex knots with metamaterials”

(Nature Communications: NCOMMS-19-41305-T)

Response to reviewer #1 comments

The paper presents theoretical and experimental realizations of acoustic knotted vortices. I think the experimental measurements of these fragile and intricate fields are enough to warrant publication. I am sure that people working on knots (being them optic, plasma, water flow...) will appreciate this achievement. The paper is nicely written, to the point and properly motivated. That is why I would recommend the paper for acceptance.

Reply: We would like to thank the referee for the positive evaluation of the present work.

However, I would like more information on how the knots were detected once that the acoustic field was scanned. In the paper, the authors talk about "dark points", I imagine that these are points at which theoretically the amplitude is 0. But, I imagine that experimentally there are some details about how to get this. Do the authors look for minima of amplitude? In any case, I am not convinced by this method. Most of the papers on knotted vortices that I have seen look in the phase and check for vortices using algorithms based on the phase. No plots of the phase appear in the main paper.

Reply: This is an important point raised by the referee. **The “dark points” in the manuscript refer to the points at which theoretically the pressure amplitude is zero, as indicated by the reviewer. Meanwhile, the dark points correspond to vortex points with phase singularities.** The experimental results of ideal “zero” pressure amplitude are not available because the background noises, though largely reduced in experimental tests, cannot be fully removed. For this reason, the experimental dark points are recognized by seeking the local minima of pressure amplitude contour. **For further verification of experimental dark points, the measured phase distributions have been added to the revised manuscript (Fig. 3), as suggested by the reviewer.** As an example, shown below are the pressure amplitude and phase distributions at the slice of $z=22\text{cm}$ of knotted acoustic fields. According to the amplitude distribution, the dark points correspond to the local minima of pressure amplitude, as marked by white dots. It is clearly seen from the phase distribution that the acoustic pressure phase undergoes one complete 2π rad cycle in a path circulating these dark points, forming the phase vortex pattern. Both sets of results and their excellent agreements show a clear demonstration of acoustic vortex knots.

Figure: Measured pressure amplitude and phase distributions at the slice of $z=22\text{cm}$ of acoustic knot

Action taken:

- In the revised manuscript, the measured phase distributions have been added to Fig. 3, and the relevant discussions have been added in the main text.
- More information on how the knotted acoustic fields were detected has been added to the revised manuscript.

Trapping of particles in acoustic vortices or shaped fields is described but no reference is provided, I would recommend these: 1. Marzo, A., Seah, S. A., Drinkwater, B. W., Sahoo, D. R., Long, B., & Subramanian, S. (2015). Holographic acoustic elements for manipulation of levitated objects. *Nature communications*, 6, 8661. 2. Baresch, D., Thomas, J. L., & Marchiano, R. (2016). Observation of a single-beam gradient force acoustical trap for elastic particles: acoustical tweezers. *Physical review letters*, 116(2), 024301.

Reply: We would like to thank the referee for the recommendation of these references, which we have cited as Ref. [46] and Ref. [47]. We also included Refs. [48, 49] as references regarding particle trapping with acoustic vortices. Additionally, in the discussion part, we added a more detailed discussion about the potential applications of particle trapping with acoustic vortex knots.

Response to reviewer #2 comments

This paper reports on theoretically and experimentally determined acoustic vortex knots (a Hopf link and Trefoil knot) using a metamaterial aperture. Design of the metamaterial was guided by theoretical formulations from topology for the particular types of vortex link and knot examined. The metamaterial aperture consists of an arrangement of 24 x 24 units (pixels), that was fabricated using SLA 3D printing. Each unit cell was numerically optimized and consisted of a coating unit cell and perforated panels that provided modulation of both the magnitude and phase. Acoustic testing was performed, and the 2D scans of sound pressure magnitude were taken at multiple distances from the metamaterial aperture, with results showing a clear demonstration of an acoustic link and acoustic knot.

This paper is well-written, and clearly and concisely describes the work presented. While vortex waves and vortex knots have been examined extensively in other fields of science (such as optics), it has to this reviewer's knowledge never been experimentally demonstrated in acoustics before. As a result, this work represents a significant and novel contribution to the scientific literature on this topic. Furthermore, the authors were thorough in their analysis, utilizing both theoretical and numerical simulation and optimization techniques, and even performing component level experiments examining the viscous effects of air on their metamaterial design (as described in the supplementary section).

Reply: We would like to thank the referee for the positive evaluation of our work.

Although overall very good, several minor issues were identified with the paper which should be addressed:

1) Given that this appears to be the first paper to demonstrate the effect of an acoustic link and acoustic knot, the authors should add some addition emphasis of their analysis and results in context of previous results in related fields, such as optics. In particular, the authors should comment on if there are any differences between their equations and measured results for acoustic (longitudinal) waves and previous results with electromagnetic (transverse) waves.

Reply: We would like to thank the referee for the helpful suggestions. More analyses and discussions in the context of previous results in related fields including optics have been added into the "discussion" section of the revised manuscript. In particular, the differences between equations, realization schemes and measured results of links and knots for acoustic (longitudinal) waves and previous results with electromagnetic (transverse) waves are highlighted, as generalized below.

In electromagnetics, the behavior of a monochromatic light beam propagating along the z -direction with three polarized components of electric fields \mathbf{E} is described by the *vector* Helmholtz equation:

$$\begin{aligned}\nabla^2 \mathbf{E}(x,y,z) + k^2 \mathbf{E}(x,y,z) &= 0, \\ \nabla \cdot \mathbf{E} = \nabla_x E_x + \nabla_y E_y + \nabla_z E_z &= 0,\end{aligned}$$

where the transverse electric field components are E_x and E_y , and the longitudinal component is E_z . For the transverse *paraxial* case, the optical vortex knots have been experimentally constructed in the laser beam [16-18, 20-23] based on the *far-field* holographic scheme, where the algebraic knot function at $z=0$ plane [$f_{Hopf}(x, y, z=0)$ or $f_{Trefoil}(x, y, z=0)$] is embedded into the waist of a Gaussian

beam. In the other case, it has been proposed theoretically that the longitudinal polarization component of light can be twisted into knotted vortex lines by *non-paraxially* focusing a subwavelength optical beam that is dominated by the *evanescent fields* [19]. The vortex knots can then be identified by examining the distribution of both intensity and phase for the light field of the longitudinal polarization. **It is worth to emphasize that, the formation of knotted vortices for either transverse or longitudinal polarized optical fields reported previously were based on the diffractive holographic scheme with phase-only hologram.** As a result, the optical hologram should possess a large number of pixels (about 300×300) in order to achieve high diffraction efficiency.

By contrast, the longitudinally polarized acoustic field characterized by a scalar pressure (P) is dominated by the *scalar* Helmholtz equation:

$$\nabla^2 P(x,y,z) + k^2 P(x,y,z) = 0 .$$

To generate the unbroken vortex knot in the transmission region adjacent to the metasurface (much easier to be detected compared with the far-field projection scheme for acoustics), the **amplitude-phase holographic scheme** is adopted, in which an algebraic knot function with $z=z_{Hopf}$ (or $z=z_{Trefoil}$) [$f_{Hopf}(x, y, z=z_{Hopf})$ or $f_{Trefoil}(x, y, z=z_{Trefoil})$] is embedded into the initial wavefront at the aperture. In contrast to the phase-only holographic scheme, each pixel of the designed acoustic metasurface can generate acoustic fields with both the prescribed amplitude and phase. In this case, **fewer number of pixels are needed** comparing with the phase-only aperture for the creation of optical vortex knots. In the present study, acoustic vortex knots are created by a compact acoustic metasurface consisting of only 24×24 pixels with the ratio of the lateral size to the operating wavelength far less than that of a phase-only optical hologram.

To summarize, in contrast to previous results in related fields, the novelty of the present work lies in that:

- The experimental demonstration for acoustic links and knots is provided for the first time.
- The present study indicates the possibility to create linked and knotted wave fields in the near-field regime using a compact aperture with fewer unit cells, which is believed to be inspirational for other physical fields.

Action taken:

- The above discussions have been added into the “Discussion” part of the revised manuscript to highlight the difference between the optical and acoustic vortex knots.

2) While the Methods section is overall very detailed, the authors should clarify if the metamaterial was 3D printed as one entire piece, or was printed in smaller sections and assembled. If it was printed in small sections, the authors should describe how they attached those different sections together.

Reply: We would like to thank the referee for noticing this point. Without influencing test results, the metamaterial sample is divided into four smaller sections of equal size. Each section is fabricated by using 3D printing, and then combined together into one entire sample.

Action taken:

- In the revised manuscript, we have added a detailed description of the sample fabrication in the method part.

3) A few minor typos were found that should be corrected: (1) In the following text from p.4: “For example, the transverse size of the sample may be up to 6mx6m for an audible frequency 6 kHz, suppose that the pixel size is of 1/3 operating wavelength. (ii) Except the fabrication trouble of bulky samples...” Here, “suppose” should be replaced with a different word, such as “given”, and “Except the” should be “Except for the”. (2) In the last paragraph on p.4, In “To enable of proposal experimentally feasible”, “enable” should be “make”. At the end of the same sentence, “24 multiplied by 24 pixels” should just read “24 x 24 pixels”. (3) In the first line of p. 6, “At the case” should be “For the case”. Once the issues listed above are addressed, I recommend that this manuscript is ready for publication in Nature Communications

Reply: We would like to thank the referee for the careful review and positive remarks about our work. Following the referee’s suggestion, the typos have been corrected in the revised manuscript.

Response to reviewer #3 comments

In this article, Zhang et al describe the creation of acoustic knots--essentially acoustic pressure holograms with knot-like 3D shapes--by means of a planar metamaterial array whose units can independently manipulate amplitude and phase. While the metamaterial design is not novel (it was introduced by Tian et al. in their 2017 letter "Acoustic holography based on composite metasurface with decoupled modulation of phase and amplitude", Ref. [42] in this article), the application to knots seems to be. The topic of this article is interesting, and it has the potential to attract the attention of others in the field. However, the fact that some concepts are not clearly explained and the absence of a powerful demonstration of the usefulness of the phenomenon being investigated prompt me to think that this article might be better suited for a technical journal rather than a high impact multidisciplinary venue like Nature Communications. The reasoning behind my statement is elaborated in the following.

Reply: We would like to thank the referee for the careful review and valuable suggestions, which have helped to greatly improve the manuscript. In the revised manuscript, some concepts have been clearly explained and a powerful demonstration of the usefulness of the phenomenon has been given. We have made all necessary changes in the manuscript by highlighting the novelty of the present study and adding more discussions on potential applications of knotted acoustic fields. We believe that the present manuscript matches the high standard and diverse readership of Nature Communications. Below please find our detailed responses to the reviewer's comments.

1) Concern on novelty. While it is very interesting that the authors were able to create knotted regions of negative pressure, I can't help but wonder whether this is something that is easy to achieve given the design in Ref. [42]. Can the authors clarify if the novelty of their work goes beyond creating a 3D hologram of a knot? Or, how is what they are doing different than the holography demonstrated in Ref. [42]?

Reply: We would like to thank the referee for the comment. The metasurface-based acoustic holograms have been extensively studied. Nonetheless, the experimental observation of acoustic knotted vortices reported in the present work denotes **a great breakthrough that goes beyond previous contribution** (previous results in optical fields as well as the acoustic hologram in [42]), as clarified below.

Firstly- **The creation of acoustic vortex is not simply the extension of the optical counterpart.** In optics, the formation of knotted vortices for transversely polarized fields are based on the *far-field* holographic scheme with **phase-only hologram**. The knotted fields are projected into the far-field by the aperture that involves a large number of pixels (about 300×300) to achieve high diffraction efficiency. If the same scheme is used by acoustic hologram, the large transverse size of the sample, which may be up to 6m×6m for an audible frequency 6 kHz, would bring great troubles in both sample fabrication and experimental characterization, as explained in the main text of the original manuscript. To solve these problems, the *near-field* holographic scheme with **amplitude-phase hologram** is adopted for the creation of acoustic vortex. It is worth to stress that **the vortex fields formed in a small and near-field zone would become more fragile and intricate, thus are more difficult to realize in contrast to the far-field scenario.** Based on this scheme, acoustic vortex knots are successfully created **in the near field** by a compact acoustic hologram consisting of only 24×24 pixels. Compared to previous results in optical vortex fields, the

present work demonstrates the possibility to create linked and knotted vortex fields in the near-field regime using a compact aperture, which is believed to be inspirational for other physical fields.

Secondly- **The hologram design and acoustic detection relevant to the fragile and intricate vortex fields represents serious challenges, which are not ever faced by previous acoustic hologram [42].** The main differences and challenges in the present study involve:

- In [42], a 2D and far-field hologram is studied in terms of numerical analyses only. **No experimental studies relevant to a 2D hologram are conducted in [42]** to demonstrate the applicability of the prototype. By contrast, in our paper, the 3D knot hologram in the near field is generated. Such near-field, 3-dimensional field sets far more stringent requirements on the metasurface design than a 2D hologram. We would like to stress that a **3D hologram is experimentally demonstrated** in the present work, making significant efforts far beyond the contributions in [42].
- In the conventional 2D hologram, the 2-dimensional information is mapped from the image plane to the hologram plane. In our study, we aim at packing the information of 3-dimensional structured acoustic field into the metasurface, the iterative optimization algorithm for holographic generation is inherently different from [42].
- The fragile and intricate vortex fields are sensitive to the amplitude and phase distribution of the aperture. Each unit cell must be carefully designed to output the exact values of both transmission amplitude and phase. To achieve this goal, the influence of the viscous loss of air must be precisely evaluated in the cell design, which is not considered in [42].
- Due to the fragility and complexity of knotted vortex fields, the detection of fragile vortex fields in experiment is not trivial. Lots of efforts need be done to evaluate the factors that influence the experimental reliability, such as the source signal spectrum and duration time, the diffraction and reflection effect by the surroundings, etc.

In contrast to previous results in related fields, the novelty of the present work lies in that:

- The experimental demonstration for acoustic links and knots is provided for the first time.
- The present study indicates the possibility to create linked and knotted wave fields in the near-field regime using a compact aperture with fewer unit cells, which is believed to be inspirational for other physical fields.

To summarize, while the cell structure in the present work is admittedly based on the prototype proposed in Ref. [42], we are convinced that the new light that our paper sheds on the first experimental observation of acoustic knotted vortices sets it apart from previous contributions. We do hope that the referee may appreciate our significant effort in addressing this challenging issue, and find our work an exciting contribution to the community.

Ref. [42] in the original manuscript (changed to [44] in the revised manuscript):

-Tian, Y. et al. Acoustic holography based on composite metasurface with decoupled modulation of phase and amplitude. Appl. Phys. Lett. 110, 191901 (2017).

Action taken:

- In the revised manuscript, selection of above discussions has been added to highlight the novelty of the present work.

2) Concern on the applicability of knots for particle manipulation. It seems to me that the biggest achievement of this article is the creation of knot-like holograms, where the knots are essentially regions of minimum pressure. But is the pressure gradient between high- and low-pressure regions enough to concentrate particles at the knot regions?

Reply: We would like to thank the referee for the comment. Yes. The biggest achievement of this article is the creation of knot-like holograms, and the vortex knots are the patterns formed by pressure minima points, referred to as the dark points in the manuscript. These dark points are surrounded by high-pressure regions, and particles could be trapped due to the presence of an acoustic pressure gradient. **To realize the particle trapping near dark points, the magnitude of the acoustic pressure gradient, which is proportional to both the wave amplitude and frequency, need to be sufficiently large.** According to the published experimental reports [1-4, as listed below, and also included as 46-49 in the revised manuscript], **the ultrasound source (up to 40kHz, driven around 15Vpp) of high operating frequency and high energy level** was usually used for achieving the particle trapping in air. In light of the above-mentioned facts, the vortex fields created at high-intensity ultrasonic environment would be able to possess enough pressure gradient to concentrate particles at the knot regions. In the present work, the experimental platform works at an audible frequency (~6 kHz) and is dedicated to the experimental demonstration of knotted acoustic fields, while not intended for the particle trapping demonstration. **The aperture design, fabrication, and acoustic testing carried out in the audible frequency regime would be different from those used for particle trapping manipulation in an ultrasonic environment.** The subject of particle trapping manipulation deserves deep and careful study, and it lies beyond the scope of the present manuscript due to its intrinsic complexity. We believe that the effort to demonstrate the particle trapping applicability deserves the right to be published in a separate paper, and would be the next step of our research.

[1] Marzo, A. *et al.* Holographic acoustic elements for manipulation of levitated objects. *Nat. Commun.* **6**, 8661 (2015).

[2] Baresch, D. *et al.* Observation of a single-beam gradient force acoustical trap for elastic particles: acoustical tweezers. *Phys. Rev. Lett.* **116**(2), 024301 (2016).

[3] Memoli, G. *et al.* Metamaterial bricks and quantization of meta-surfaces. *Nat. Commun.* **8**, 14608 (2017).

[4] Ozcelik, A. *et. al.* Acoustic tweezers for life sciences, *Nat. methods* **15**, 1021-1028 (2018).

From [Memoli *et al.*, Metamaterial bricks and quantization of meta-surfaces, Nature Communications, 2017], particle trapping can occur in low amplitude pockets surrounded by high pressure regions. But, in the current work, the low pressure regions seem to be distributed and wide, rather than highly localized. Do the authors have any insight in this?

Reply: This is an important point raised by the referee. While the referenced paper [Memoli *et al.*, NC, 2017] demonstrated the particle trapping by using acoustic nodes, i.e., low amplitude pockets surrounded by high pressure regions, there were published literature reporting that **acoustic node lines, as an extension of acoustic nodes, have been proven favorable for particle trapping manipulation** [Proceedings of the National Academy of Sciences (2017), 114(40), 10584-10589; Small (2020): 1906394; Advanced Materials 30.43 (2018): 1802649]. The low-pressure region of acoustic vortex fields realized in our paper is essentially the curved node line pattern; Therefore, it

holds the potential for particle trapping manipulation. Distinct from the previous node line pattern formed by standing waves, **the present work demonstrates the 3D topology configuration of curved and knotted node lines**. The particle trapping strategy based on this novel knotted node line is expected to have an impact on a wide range of acoustic tweezer applications by unlocking the possibility of patterning biological and chemical particles to form structures in the 3D space.

Action taken:

- In the revised manuscript, we have added a more detailed discussion of the potential and limitations of the acoustic knots for trapping applications in the discussion part.
- In the revised manuscript, we included Refs. [46-52] as references regarding particle trapping with acoustic vortices.

3) The article seems scientifically sound, but it lacks a field-opening demonstration that would justify publication in a high-profile journal like Nature Communications. Have the authors tried to demonstrate that their knots can indeed be used for complex 3D particle manipulation? In my opinion, a demonstration of this type would raise the article to a much higher level. Just speculating on the potential applicability of these knots, in my opinion, is not enough

Reply: We would like to thank the referee for the comment. We agree that a demonstration of the novel 3D particle trapping with knotted acoustic fields is of great significance. However, to realize the particle trapping near dark points, the magnitude of the acoustic pressure gradient, which is proportional to both the wave amplitude and frequency, need to be sufficiently large. According to the published experiments [1-4], the ultrasound source of high operating frequency and high energy level (up to 40kHz, driven around 15Vpp) was usually used for achieving the particle trapping in air. In the present work, the experimental platform works at an audible frequency (~6 kHz) and is dedicated to the experimental demonstration of knotted acoustic fields, while not intended for the particle trapping demonstration. **The aperture design, fabrication, and acoustic testing carried out in the audible frequency regime would be different from those used for particle trapping manipulation in an ultrasonic environment.** We do understand the referee's expectation of the potential applicability of acoustic knots. In the revised manuscript, we have provided a more detailed discussion about the potential and limitations of particle-trapping application. The particle-trapping demonstration deserves deep and careful study. But currently, it lies beyond the scope of the present manuscript due to its intrinsic complexity. We believe that the effort to demonstrate the particle trapping applicability deserves the right to be published in a separate paper, and would be the next step of our research.

[1] Marzo, A. *et al.* Holographic acoustic elements for manipulation of levitated objects. *Nat. Commun.* **6**, 8661 (2015).

[2] Baresch, D. *et al.* Observation of a single-beam gradient force acoustical trap for elastic particles: acoustical tweezers. *Phys. Rev. Lett.* **116**(2), 024301 (2016).

[3] Memoli, G. *et al.* Metamaterial bricks and quantization of meta-surfaces. *Nat. Commun.* **8**, 14608 (2017).

[4] Ozcelik, A. *et. al.* Acoustic tweezers for life sciences, *Nat. methods* **15**, 1021-1028 (2018).

4) About topological protection of the knotted shapes. The authors mention in several points of the article that their knots are topologically-protected, and this is the reason why 24x24 cells are enough to see knot-like shapes. I found this argument very hand-wavy, and unsatisfactory. How does topology play a role here? If you lower the number of cells below 24x24, do you still see knotted shapes owing to this topological protection? When would this protection cease to be effective? I urge the authors to be clearer, and more rigorous, about this.

Reply: We would like to thank the referee for indicating this point. The mathematical knots are knotted lines whose ends are joined together so that they cannot be undone. In topology, two knots are equivalent if one can be transformed into the other via the continuous deformation without cutting the lines or permitting the lines to pass through itself [3-11]. In our study, it is found that the knotted shape of the acoustic vortex line, despite being deformed and distorted as the aperture's pixel number decreases, would maintain topologically invariant. For a clear illustration, numerical simulations are conducted to calculate the vortex lines created by apertures with different sets of pixel numbers: 20×20, 14×14, 12×12, 10×10 for the Hopf link, and 20×20, 16×16, 12×12, 8×8 for the Trefoil knot, as shown in Figs. S1 and S2 of the supporting information, respectively. It can be clearly observed that **the Hopf link and Trefoil knot vortex lines deform in size and shape with the decreasing of the cell number, yet maintain topologically invariant.** According to numerical analyses, the minimum number of pixels are found as 12×12 for links and 16×16 for knots; Below this lower limit, the topological phase transition of the vortex line (broken and reconnection) would appear such that the knot and link are destroyed, and the topological protection ceases to be effective. According to above analyses, a moderate pixel number 24×24 above this lower limit is chosen, which is adequate to our needs for the vortex knot demonstration.

Action taken:

- In the revised manuscript, we have added the discussion on pages 4 and 5 to clearly illustrate the topological protection of the knotted shape.
- In the revised manuscript, numerical results of acoustic fields generated by apertures with different sets of pixel numbers are added in Figs. S1 and S2 of the supporting information.

5) Level of detail. Most explanations in the article are not very clear; this stems from the fact that the authors do not provide a sufficient level of detail in the main article. Sure, the Supplementary Material (SM) helps. But the SM should not be used as a place to dump all the explanations, leaving the main article to become a superficial treatment of the topic.

Reply: We would like to thank the referee for raising the following important suggestions, which have helped to greatly improve the manuscript. More detailed discussions have been added to the revised manuscript according to the reviewer's suggestions, as addressed below.

-For example, the authors talk about optimization to design the metamaterial with 24x24 cells. But how is this optimization done? Why did the authors choose 24x24 cells? What happens if they choose more or less? This is not even clear from the SM.

Reply: We would like to thank the reviewer for noticing this point. The optimization process for choosing the 24×24 cells is explained as follows. In principle, the larger the number of pixels is, the better is the knotted vortex performance. To facilitate the sample fabrication and testing, the

optimization goal is to determine the minimum cell numbers required. To this end, numerical simulations are conducted to calculate the vortex lines created by apertures with different sets of cell number: 20×20 , 14×14 , 12×12 , 10×10 for the Hopf link, and 20×20 , 16×16 , 12×12 , 8×8 for the Trefoil knot, as shown in Figs. S1 and S2 of the supporting information, respectively. As the cell number decreases, the Hopf link and Trefoil knot vortex lines deform in size and shape, yet maintain topologically invariant. According to numerical analyses, the minimum number of pixels are found as 12×12 for links and 16×16 for knots; Below this lower limit, the topological phase transition of the vortex line (broken and reconnection) would appear such that the knot and link are destroyed. In our study, a moderate pixel number 24×24 that lies above the lower limit is chosen, which is adequate to our needs for the vortex knot demonstration.

Action taken:

- In the revised manuscript, we have added the discussion on pages 4 and 5 to illustrate the pixel design in the metasurface.
- In the revised manuscript, we have added the numerical results of acoustic fields generated by the aperture with different pixel numbers in Figs. S1 and S2 of the supporting information.

-The geometry of a unit cell is described in the caption of Fig.1, and the authors mention the pipeline length L in both the main text and caption. However, from the figure, it is not really clear what L is. It is also not clear if all that matters is the path length or its shape.

Reply: We would like to thank the reviewer for noticing this point. The pipeline length L refers to the path length along the bent cavity. The length " L " has been marked in Fig. 1e of the revised manuscript.

-Why do the authors operate the metamaterial at a frequency of 6 kHz? There is absolutely no explanation on this point in the article. Did they choose the frequency arbitrarily? This demonstrates that the treatment is superficial and that the work is hardly reproducible the way it is at the moment.

Reply: We would like to thank the referee for noticing this point. The operating frequency of 6 kHz is not chosen arbitrarily. Although the proposed scheme for the creation of acoustic vortex knots is not limited to a specific range of frequencies, there may exist an optimum frequency for vortex field demonstration in the experiment due to the restriction of sample fabrication and testing system.

The acoustic vortex field occupies the larger space as the operating frequency decreases. In our tests, the maximum measurement area of the scanning stage is $50\text{cm} \times 50\text{cm}$. This transverse size would set a lower bound for the operating frequency. On the other hand, a higher frequency results in a smaller cell size of the sample. Due to the limitation in precision by the 3D printing technique, there exists an upper bound for the operating frequency in order to ensure a high-quality fabrication of the cell structure with complex geometry. In addition, the vortex field would be compressed into a smaller space at a higher frequency. In this case, sound measurement in a small space would be more difficult. With a balance about the above factors, the operating frequency of 6 kHz is chosen in our study.

Action taken:

- In the revised manuscript, we have added the discussion on page 8 to illustrate the reason for choosing the operation frequency 6kHz.

-How are the 3D reconstructions of the knots in Fig.3 obtained? Are the authors tracking the minima of the pressure slices shown in the same figure? It seems to me that these pressure maps present entire regions with pressure close to 0. So, how do the authors select only two or four points per slice? Are they local minima? This is absolutely not clear from the text. Also, how is the trefoil knot colored in Fig.3b?

Reply: We would like to thank the referee for the very important comment. The knots are patterns formed by points with zero pressure amplitude, referred to as “dark points” in the manuscript. The experimental results of ideal “zero” pressure amplitude are not available because the background noises, though largely reduced in testing, cannot be fully removed. For this reason, the experimental dark points are recognized by seeking the local minima of pressure amplitude contour. Based on the fact that the dark points are also vortex points with phase singularities, **we have provided further verification of 3D reconstructions of the knots in the revised manuscript by examining the phase pattern of acoustic pressure.** The measured phase distribution at each slice has been added to Fig. 3 of the main text. As an example, shown below are the pressure amplitude and phase distributions at the slice of $z=22\text{cm}$ of knotted acoustic fields. According to the amplitude distribution, the dark points correspond to the local minima of pressure amplitude, as marked by white dots. It is clearly seen from the phase distribution that the acoustic pressure phase undergoes one complete 2π rad cycle in a path circulating these dark points, forming the phase vortex pattern. Both sets of results and their excellent agreements show a clear demonstration of acoustic vortex knots.

The “rainbow” coloring of the trefoil knot in Fig. 3e,f is just used to visualize the knots clearly and distinguish the over-strand from the under-strand at each crossing. A statement has been added to the caption of Fig. 3 to explain what the color of the trefoil knot means.

Figure: Measured pressure amplitude and phase distributions at the slice of $z=22\text{cm}$ of acoustic knot

Action taken:

- In the revised manuscript, we have added the results of the measured phase distribution in Fig. 3.
- In the revised manuscript, we have added the description on page 8 to illustrate the phase distribution around the dark point.

- In the revised manuscript, we have added an explanation in the caption of Fig. 3 to illustrate the meaning of color on the knotted line.

Minor suggestions

-The work is not badly written, but the article "the" is incorrectly used in various parts of the text. This should be fixed.

Reply: We would like to thank the referee for the useful comment. We have polished the revised manuscript and corrected the usage of the word “the”.

-The authors, in the main text, comment on the partial differences between numerical and experimental knots. However, these differences are very hard to see, since the two sets of knots are not plotted next to each other. I think the authors should add the numerical 3D knots to Fig.3, and plot both numerical and experimental knots from an identical viewpoint. Only then, it would be possible for the reader to compare them directly.

Reply: We would like to thank the referee for the helpful suggestion. In the revised manuscript, both numerical and experimental knots are shown from an identical viewpoint in Fig. 3 for direct comparison between them.

Reviewers' Comments:

Reviewer #1:

Remarks to the Author:

The authors have answered all my questions and applied the requested changes. Therefore, I recommend this paper for acceptance.

Reviewer #2:

Remarks to the Author:

This reviewer appreciates the thorough responses and additional content that the authors have provided in their revised manuscript, and believes that the authors have addressed all of the concerns brought up in the initial review. As a result, I recommend that this manuscript is ready for publication in Nature Communications.

Reviewer #3:

Remarks to the Author:

I am satisfied with the way the authors have addressed my comments. I do recognize that no one has demonstrated the creation of 3D acoustic holograms in air using such a compact platform (as it is also clear from a very recent review on the topic: <https://doi.org/10.1063/1.5132629>).

Prior to publication, I think the article should be edited to fix some clunky sentences and grammatical errors. For example, in the new "Discussion", the authors have basically copy-pasted the comments given as responses to the authors; I expect the authors to do a better job and create a cohesive paragraph that reads well. For example, saying "This subject deserves deep and careful study and would be the next step of our research" in the article is not needed; it makes sense as a response to a reviewer but not in an article.

Response to reviewer #1 comments

The authors have answered all my questions and applied the requested changes. Therefore, I recommend this paper for acceptance.

Reply: We thank reviewer #1 for the recommendation.

Response to reviewer #2 comments

This reviewer appreciates the thorough responses and additional content that the authors have provided in their revised manuscript, and believes that the authors have addressed all of the concerns brought up in the initial review. As a result, I recommend that this manuscript is ready for publication in Nature Communications.

Reply: We thank reviewer #2 for the recommendation.

Response to reviewer #3 comments

I am satisfied with the way the authors have addressed my comments. I do recognize that no one has demonstrated the creation of 3D acoustic holograms in air using such a compact platform (as it is also clear from a very recent review on the topic: <https://doi.org/10.1063/1.5132629>).

Reply: We thank reviewer #3 for the recommendation.

Prior to publication, I think the article should be edited to fix some clunky sentences and grammatical errors. For example, in the new "Discussion", the authors have basically copy-pasted the comments given as responses to the authors; I expect the authors to do a better job and create a cohesive paragraph that reads well. For example, saying "This subject deserves deep and careful study and would be the next step of our research" in the article is not needed; it makes sense as a response to a reviewer but not in an article.

Reply: According to the reviewer's suggestion, we have edited the article carefully to fix some clunky sentences and grammatical errors.